# Adherence to CPAP Treatment: Can Mindfulness Play a Role?

**DOI:** 10.3390/life13020296

**Published:** 2023-01-20

**Authors:** Athanasia Pataka, Seraphim Chrysovalantis Kotoulas, Panagiotis Raphael Gavrilis, Alice Karkala, Asterios Tzinas, Aimiliza Stefanidou

**Affiliations:** 1Respiratory Failure Unit, G. Papanikolaou Hospital Thessaloniki, Aristotle University of Thessaloniki, 57010 Thessaloniki, Greece; 2ICU, Hippokratio General Hospital, 54942 Thessaloniki, Greece; 3Medical School, Aristotle University of Thessaloniki, 54124 Thessaloniki, Greece; 4American College of Thessaloniki (ACT), Vasiliou Sevenidi 17, Pilea, 55535 Thessaloniki, Greece

**Keywords:** obstructive sleep apnea, OSA, CPAP, adherence, mindfulness, cognitive therapy

## Abstract

Obstructive sleep apnea (OSA) is considered a chronic disease that requires long-term multidisciplinary management for effective treatment. Continuous Positive Airway Pressure (CPAP) is still considered the gold standard of therapy. However, CPAP effectiveness is limited due to poor patients’ adherence, as almost 50% of patients discontinue treatment after a year. Several interventions have been used in order to increase CPAP adherence. Mindfulness-based therapies have been applied in other sleep disorders such as insomnia but little evidence exists for their application on OSA patients. This review aims to focus on the current data on whether mindfulness interventions may be used in order to increase CPAP adherence and improve the sleep quality of OSA patients. Even though controlled trials of mindfulness and CPAP compliance remain to be performed, this review supports the hypothesis that mindfulness may be used as an adjunct method in order to increase CPAP adherence in OSA patients.

## 1. Introduction

Obstructive sleep apnea (OSA) is the most prevalent sleep breathing disorder (SDB) caused by complete or partial upper airway occlusion during sleep. The incidence of OSA is higher than previously believed, with almost 20% of adult males and 10% of women suffering from the moderate-to-severe disease. Obesity is considered to be one of the most important risk factors for OSA [1]. Apart from obesity, increased neck circumference, male sex, older age, upper airway, and craniofacial abnormalities are also considered significant clinical risk factors of the disease [2]. Several cluster analysis studies have found that the classical phenotype of OSA, i.e., the obese sleepy male, represents only a part of the patients, and have identified other clinical phenotypes, with atypical symptoms, such as insomnia, gender-specific, with different co-morbidities and polysomnographic findings [3].

OSA interrupts the physiological sleep structure leading in sleep fragmentation causing excessive daytime sleepiness (EDS), and impaired vigilance, resulting in an increased risk of work and motor vehicle accidents [4]. Undiagnosed and untreated OSA has been associated with increased mortality and has serious health consequences such as hypertension, arrythmias, cardiovascular and cerebrovascular disease, diabetes, impairment of heart failure, and pulmonary hypertension [5]. All the aforementioned consequences have a significant economic burden [6]. However, with early identification and treatment, the negative implications of OSA can be significantly decreased.

OSA is considered a chronic disease that requires long-term multidisciplinary management for effective treatment. Continuous Positive Airway Pressure (CPAP) is still considered the gold standard of therapy, even though there are several treatment options such as mandibular advancement devices, weight loss, lifestyle interventions, positional therapy, hypoglossal nerve stimulation, and surgical operations. CPAP is recommended as the first choice for patients with moderate to severe disease and those with mild and clinical symptoms, such as EDS, or co-morbidities [7]. CPAP provides a stream of pressurized air constantly during inspiration and expiration in order to maintain the upper airways open. The application of CPAP resolves obstructive respiratory events, improves oxygen desaturations, resulting in improved daytime sleepiness, cognitive function, and mood [8]. Additionally, treatment with CPAP has beneficial cardiovascular effects as it reduces arterial blood pressure, especially in patients with severe disease; it improves pulmonary hypertension and left ventricular ejection fraction in patients with heart failure [9,10]. CPAP also reduces mortality and improves the quality of life [11,12,13,14]. A dose-response relationship has been found between the improvement in health and CPAP adherence [9,11,12,13,14]. However, CPAP effectiveness is limited due to poor patients adherence. It has been shown that almost 50% of patients discontinue CPAP after a year of treatment [15,16,17], within a range of 29% to 83%, while 8 to 15% of patients reject treatment as early as the first night of application [18,19]. In the comprehensive systematic literature review of Rotenberg et al. [18] that evaluated data from 82 trials regarding CPAP adherence over a twenty-year timeframe, it was found that CPAP adherence remained persistently low, around 34% (30–40%). It was also found that approximately 11% of the participants of the trials were unable to remain on CPAP treatment over the duration of the trial. Discontinuance of CPAP is a global problem despite the different cultural characteristics of the patients.

Mindfulness is defined as being in the moment and aware of one’s thoughts and emotions. Living in the present moment through mindfulness practices can be a state or a trait characteristic that is an important element for a healthy life [20,21,22]. Mindfulness-based therapies have been used in order to improve insomnia and sleep quality, especially in individuals who prefer these types of therapies and those with an expectation of benefit [23]. However, few studies have focused on the effect of mindfulness on OSA treatment, especially as an intervention to increase CPAP adherence and controlled trials remain to be performed. The current review focuses on the evaluation of the hypothesis of whether mindfulness may be used as an alternative method in order to increase CPAP adherence.

## 2. Practical-Technical Issues Affecting CPAP Adherence

Adherence to CPAP is important for OSA patients. The duration of CPAP use that is required in order to normalize functioning is still unclear, ranging from at least 4 h [24,25] and reaching up to 6–8 h per night of >70% of nights (i.e., >5 nights/week) in different studies [26,27]. Non-adherence to CPAP treatment is attributed to multiple factors including disease severity, possible side effects during the application of the device, psychological factors, and socio-demographic/economic characteristics of the patient [28,29]. The rate of CPAP adherence has been affected by several barriers to successful treatment as mask leaks, skin irritation, conjunctivitis, nasal congestion, dry throat, claustrophobia, or aerophagia [29] (Table 1). CPAP use within the first week is predictive of long-term use [30,31]. For that, it is crucial for the treating physician to assess the possible risk factors for non-adherence early in the application of a treatment, preferably within the first 2 weeks of use [32]. It has been shown that healthcare professionals have different perceptions and knowledge compared with patients regarding CPAP side effects, possible problems, and educational needs. This is important in the design of educational programmes for healthcare professionals and patients in order to increase CPAP adherence [33]. The education that is provided by a knowledgeable and trusted health professional regarding the use of the CPAP device and its expected benefits is important.

## 3. Other Variables Affecting CPAP Acceptance

As the pattern of CPAP adherence is evident during the first weeks of treatment, it may be hypothesized that the patient has already formed perceptions regarding OSA and possible treatment benefits. Based on this hypothesis, the most effective methods in order to promote adherence are based on the perception of the patient [34]. Education is recommended by the American Academy of Sleep Medicine as an important component of adherence [35]. However, there is evidence those educational interventions about OSA and its consequences when untreated, on the different types of devices and masks, or on providing solutions to resolve the different problems, did not result in the complete improvement of adherence [16,17,31,32,33]. Therefore, adherence to CPAP may depend on other factors such as environmental, motivational, and psychological, and not only technological.

Apart from the various practical issues which have often been considered barriers to treatment adherence (Table 1), psychological variables have also been examined in the prediction of CPAP acceptance. Understanding and assessing patients’ prior beliefs regarding their expectations of health care is very important and highly relevant to overall patient satisfaction. According to the Health Belief Model (HBM), it is crucial to communicate effectively to the patients any practical issues and negative experiences which may impact health outcomes [36]. Addressing patient preferences and factors that may cause discomfort across various clinical contexts may achieve a greater likelihood of adherence and avoid intense emotions such as love or hate which are often associated with CPAP use [37]. Patients report a major subjective improvement even the first morning after treatment which could be viewed as an initial quality indicator for future research [38,39].

Patients’ level of adherence pattern is strongly associated with their expectations and beliefs [37]. The use of CPAP significantly decreases over 12 months and the decline may be predicted by the experiences of patients with the device early (i.e., at 1 month), making intensive early interventions more feasible to improve long-term compliance [38,39]. Understanding how a person meets or resists expectations could help clinicians identify how some patients adhere to treatment plans and others don’t, based on their ‘inner’ and ‘outer’ expectations. While most people are not exclusively inner-driven or outer-driven, our tendency influences our behavior. Taking into account the particular psychological factor in understanding CPAP adherence could provide a more holistic framework for offering tailored therapeutic interventions and improving patient engagement [40].

Psychological well-being has also been reported to be severely affected in individuals suffering from OSA and obesity. A core aspect of mental health, psychological well-being, has also been evidenced to be deeply affected in OSA patients who are obese. The study by Scarpina et al. was the first to document the role of OSA in the subjective perception of psychological well-being [41]. In a similar psychosocial direction, social processes, such as personal perception and close relationships should be examined. Sleep may be universal but there are variations in the social context it occurs and social psychological research should investigate such social and cultural disparities. Sleep deprivation has been shown to impair cognitive functioning and heuristic tendencies leading to stereotyping and bias. Bodenhausen has examined the role of stereotypes and biases, especially in a lack of motivation, and has found that “morning people” were more likely to engage in stereotyping at night and “night people” were more likely to engage in stereotyping in the morning [42]. The importance of circadian variations is particularly interesting because they may affect several different processes, i.e., biological, social, and cognitive. The knowledge of this may provide a different perspective on the expectations of CPAP compliance according to circadian variations. It would be rather difficult to expect a ‘night’ person to apply CPAP early at night than a ‘morning’ person.

Recent studies have also shown that sleep deprivation impairs empathic responding and overall social experience with reference to interpersonal relationships. Sleep-deprived partners who are not well-rested report more interpersonal conflicts and low frustration tolerance which may set them on a path to an unhappy relationship [43,44]. Beyond any relationship problems, poor sleep can take other social dimensions, such as making patients and partners feel lonely, detached, and eventually withdrawn [22,44]. The emotional interdependence of sleep partners highlights the social component of sleep, and how partners of OSA patients are integral elements contributing to any successful intervention. Marital quality and partner involvement affect adherence to CPAP and has been identified as an important research need to be addressed [45]. Examining the relationship between partner dynamics and sleep, OSA creates a collateral burden to spouses and/or bedtime partners who may often sleep apart from their partners suffering from OSA [46,47]. Partners frequently complain about snoring and sleep interruptions but are also worried about their bed partners who experience various breathing abnormalities during the night [48]. In conclusion, the consequences of OSA itself and its treatment expand beyond the individual that suffers from the disease [45].

Recent research has shown that a patient’s personality traits, such as health locus of control and low self-efficacy are other factors for non-adherence. Patients who have a strong internal locus of control and assume personal responsibility for their health are more likely to adhere to treatment [49]. These patients believe that they can change the situation when needed, they are more receptive to consult or following medical advice and are more self-efficacious [50]. Highly empowered patients perceive their condition as urgent, want to take control of their own health, and are more motivated to start CPAP therapy in the first place (self-referring is common), and are more likely to adhere to treatment [50]. On the other hand, patients with Type D personalities are reluctant to follow medical advice and show decreased adherence to CPAP treatment [51]. Individuals with type D personality (D stands for distressed) are characterized by increased negative emotions in different situations and social inhibition as they do not share their emotions with others, because they are afraid of possible disapproval or rejection. Future studies should focus on investigating the causes of low adherence and construct a specific protocol on the basis of personality characteristics and co-morbidity.

Adherence to CPAP may also be influenced by various psychological conditions, especially claustrophobia. Claustrophobia is a type of an anxiety disorder where one can experience anxiety when in a confined space. Claustrophobia is not the same for everyone, it can range from mild anxiety to a panic attack but it is perceived by many patients as one of the most significant obstacles to CPAP therapy [52]. Because of the overall inconvenience and discomfort of the mask and the tube piece, claustrophobia is a common reaction with patients experiencing shortness of breath and feelings of suffocation [52].

## 4. Interventions to Improve Non-Adherence to CPAP Therapy

During the last decades, several non-pharmacological treatments have been developed in order to help patients with sleep disorders. Sleep medicine combines the work of many health professionals, such as pulmonologists, neurologists, psychiatrists, otolaryngologists, maxillofacial surgeons and psychologists. Due to the multidisciplinary nature of sleep medicine different specialties are required to work together for the effective diagnosis and treatment. Psychology and Sleep medicine are closely related. Non pharmacologic treatment options include cognitive, behavioral, psychosocial, and educational interventions that may help in improving patients’ quality of life. In order to improve adherence to CPAP many different interventions have been used (Table 2) [40,49,53]. Behavioral sleep specialists use evidenced-based therapies combining cognitive techniques with behavioral approaches [53]. Cognitive-behavioral treatment is one of the most important behavior change interventions. A recent meta-analysis revealed that motivational interventions were more successful than educational programs and usual care in improving CPAP adherence, even though the results were not always sustained across all the studies [54].

The clinical observation that even though the therapeutic value of CPAP is undeniable, the percentage of patients that are compliant with treatment is rather low, created the need for educational and behavioral support. Despite the significant technological improvement of masks and devices and telemedicine applications, adherence to CPAP continues to be a major problem [55]. Some patients underestimate the severity of their disease due to its chronicity or some other perceives it as a disability and for that refuse treatment. The continuity of use affects compliance. When used as indicated, CPAP normalizes sleep architecture, reduces daytime sleepiness, cardiovascular risk, and improves health outcomes [56].

One of the most difficult problems to solve is the psychological acceptance of the device. Behavioral change is an important aspect in the acceptance of every treatment and is a complex procedure including not only psychological and motivational, but also socio-environmental aspects [57]. It includes the evaluation of the patient’s adherence to a treatment considering the level of awareness of the disease and its health consequences (reasons for change), the eagerness of the patient to change, the readiness of the patient to change, the perceived significance of this change and the spirit in the ability to change [58]. Several behavior change interventions have been used in order to improve adherence to treatment in several chronic conditions including respiratory disease [59] and more specifically for CPAP treatment [31,34]. The most successful intervention over the years for optimizing adherence has been behavioral therapy [60]. The comprehensive explanation to the patient and partner regarding the sleep disorder, its therapy with the function of equipment (mask, humidifier), the early resolution of problems—side effects (Table 1), psychological consultations, and a careful follow-up are the main elements that may increase the compliance [61] (Table 3).

Behavioral interventions, such as the use of cognitive-behavioral therapy (CBT) and of motivational enhancement therapy (MET) in order to increase the self-efficacy of the patient, in addition to education, seem to be a promising approach [62]. The goal of CBT is, through the conversational exchange, to correct the patients’ beliefs that are incorrect in order to change their behaviors toward treatment [63]. MET applies motivational interviewing through directed interview questions in order to reinforce patients’ motivations [64]. A comprehensive program should ideally be multifactorial including the intervention of different specialists such as sleep physicians, technologists, sleep psychologists, and nurses but also partners or caregivers.

OSA and insomnia often coexist. OSA patients present a higher prevalence of insomnia symptoms (40–60%) compared to that of the general population and this has led to the identification of a new disorder named co-morbid insomnia and OSA (COMISA), that has been highly underestimated [65]. The treatment of COMISA should combine positive-airway pressure (PAP) for OSA, together with CBT for insomnia. The combined treatment has been found to have a better patient outcome in comparison to that of every single treatment alone [65].

## 5. Mindfulness Interventions to Increase CPAP Adherence

Mindfulness, as a quite heterogeneous term in contemporary psychology, is viewed as an umbrella term that can refer to various facets of mindfulness, from a mental state to a personality trait and from a meditation practice to a type of clinical intervention. Mindfulness has been used as a form of meditation emphasizing a nonjudgmental state of complete or heightened awareness of one’s thoughts, experiences, or emotions [20,21]. Conceptualizing mindfulness as an art or as a science makes it unique in some way and different backgrounds, disciplines, ideologies, and practices try to achieve ‘ownership’ of that complicated concept. Depending on the viewing angle, mindfulness can be viewed as a ‘state’ or ‘trait’ mindfulness, but it is characterized as both since the practice of mindfulness is linked with the state and trait changes. People may change drastically during their lifetime when experiencing the benefits of mindfulness. It is worth noting that ‘state’ mindfulness can occur during meditation practices and ‘trait’ mindfulness is an individual trait that has been associated with being more conscious and aware in everyday life. ‘Trait’ mindfulness (or sometimes called ‘dispositional’ mindfulness) can be accessed through several psychometric questionnaires, such as the Mindful Awareness Scale (MAAS) and the Five Facet Mindfulness Questionnaire (FFMQ) and the Cognitive and Affective Mindfulness Scale-Revised (CAMS-R). Mindfulness skills (integration of knowledge and practice) are powerful mind/body life skills that can be applied to a variety of settings and conditions, alleviating the burden of symptoms and increasing psychological well-being [66,67,68].

A growing body of literature suggests that adding acceptance-based therapies in mindfulness approaches can optimize patient engagement and response to treatment. The idea is for the patient to accept thoughts and feelings (positive or negative) which eventually leads to self-care, a major determinant of outcomes. Mindfulness helps people to accept their experiences and become more compassionate with themselves (self-compassion) and with others as evidenced by enhanced prefrontal activation in imaging studies as fMRI and electrophysiologically in EEG [67,68]. The most widespread protocol used both in the clinical and non-clinical context is the Mindfulness-Based Stress Reduction (MBSR), a rigorous 8-week program that involves formal and informal meditation practices and was originally designed for stress reduction [20,21,68,69,70,71,72,73]. The aim of MBSR programs is to enhance well-being and coping with stress in diverse populations. MBSR has been proven to address chronic pain, depression, anxiety, and other conditions and overall increase the patient’s quality of life yielding significant benefits both in clinical and non-clinical samples [67,70,71,72,73].

Mindfulness therapies have been applied to patients suffering from sleep disorders [67,68,69]. Mindfulness interventions are suggested as a therapeutic option by the American Academy of Sleep Medicine in patients with insomnia, more frequently in a group format [69]. In this group of patients, mindfulness techniques may be also combined with other therapies, such as CBT (sleep restriction therapy, stimulus control, and sleep hygiene) [67,69]. Claustrophobia is highly prevalent among CPAP-treated patients influencing short and longer-term CPAP non-adherence [52]. In an attempt to examine if mindfulness interventions may be effective in improving CPAP adherence of OSA patients, Gawrysiak et al. [71] have structured a detailed protocol targeting claustrophobia (Mindfulness-based Exposure for PAP-associated Claustrophobia, MBE-PC) once per week for eight consecutive weeks in group meetings. The results of this study have not been published yet.

Studies demonstrate that depression, anxiety, and cognitive functions are considered complications of OSA and may be improved after using CPAP [74,75,76,77]. For that someone may consider that possibly other treatment interventions targeting psychological distress may be effective in OSA patients [78,79]. Li et al. [80] have evaluated whether mindfulness was associated with CPAP adherence using the MAAS. The authors have concluded that only MAAS and OSA severity were associated with CPAP adherence irrespective of the presence of psychological distress assessed by the Hospital Anxiety and Depression Scale (HADS); even though HADS evaluating depression was found higher in the nonadherent group.

Furthermore, chronic stress can reduce the prefrontal cortex and increase the size of the amygdala making the brain more receptive to stress. Chronic stress can also weaken emotion regulation [81]. Emotion regulation or emotional self-regulation refers to a person’s ability to affect one’s emotional state. A recent study [82] indicated that fragmented sleep and the reduction of REM sleep, which both characterize the sleep architecture of OSA, were associated with the difficulty of patients to recall details from the past and overall, with poor memory consolidation. In this regard, an embodied emotion regulation framework could be employed to understand how mindfulness, through top-down or bottom-up pathways affects emotion regulation from a cognitive or clinical perspective [79].

In addition, emerging evidence suggests that mobile Health interventions may improve treatment adherence and outcomes. Technological advancements in the digital realm can indeed improve patient compliance. Some mindfulness-related apps have been evaluated for clinical efficacy (e.g., Calm app is one app that specializes in audio and video programs intended to help someone relax before bedtime) and could be a viable option to help patients with OSA reduce self-reported anxiety and get a high-quality sleep [83,84].

## 6. Conclusions

In order for CPAP therapy to be effective, the patient needs to be committed to treatment using the device every night (or more than 5 nights/weeks) for more than 4 hours/night. The discontinuity of the appropriate CPAP use is reflected in the reduction of the amelioration of symptoms, resulting in a lesser benefit. CPAP treatment is behaviorally based and requires a multidimensional approach. This long-term commitment to treatment regarding CPAP adherence is critical. Technological, educational and behavioral, strategies may be needed in order to target the different disease and patient characteristics, and possible side effects. Personalized medicine should be the future target of treatment individualizing adherence goals by treating patient-specific symptoms (such as excessive daytime sleepiness or insomnia) and reducing the risk of patient-specific consequences (such as cardiovascular) [3]. This review supports the hypothesis that mindfulness can therefore serve as a novel approach to promote CPAP adherence in OSA patients by reducing emotional distress and increasing subjective well-being (Figure 1). As controlled trials have not been performed yet, future research should continue to investigate the role of mindfulness-based interventions in CPAP treatment adherence

## Figures and Tables

**Figure 1 life-13-00296-f001:**
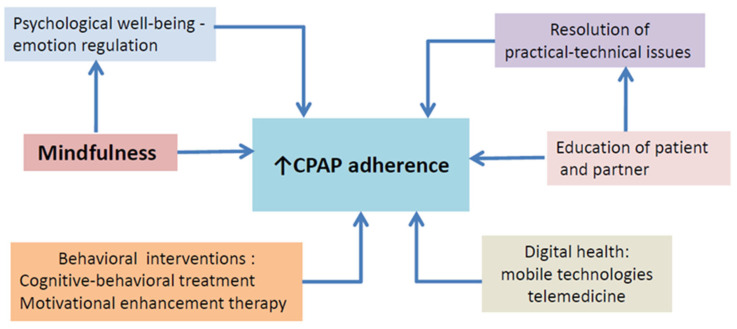
Summary of the different strategies used to increase CPAP adherence. CPAP = Continuous Airway Pressure.

**Table 1 life-13-00296-t001:** Problems—practical issues during CPAP use and their solution.

Problem	Solution
Leaks	Better fitting of the mask For the nose mask, use a chinstrap for mouth leaks Try different mask
Skin lesions	Better fitting of the mask Try different mask Topical application of products for skin issues
Rhinitis	If existed previously: increase treatment (inhaled steroids, antihistamines, very short course of oral steroids) If it did not exist previously: examine for possible leaks, examine for persistence of symptoms, possible allergic test and rhinomanometry 2 weeks with inhaled steroids, antihistamines, and/or ipratropium bromide Check again for adequate mask fit (use chinstraps) If no improvement: humidification If no improvement: ENT referral, change to oronasal mask
Conjunctivitis	Better fitting of the mask Try different mask
Dry mouth	Better fitting of the mask For nose mask, use a chinstrap for mouth leaks Try oronasal mask Humidification
Noise	Better fitting of the mask
Aerophagia	Better fitting of the mask A transient problem usually
Removal during the night involuntary	Better fitting of the mask to avoid leaks Try different mask Explain that nothing will happen Set an alarm clock in order to put on the mask
Cold air	Humidification
Claustrophobia	Select smaller interfaces such as a nasal pillows or nasal masks Wear CPAP while awake and practice breathing through the mask during the day while reading a book, watching TV Gradually increasing the time of use Select Ramp facility and Expiratory Pressure Relief
Anxiety, phobia, negative social aspects	Psychotherapy Enhance self-efficacy

CPAP: Continuous Positive Airway Pressure.

**Table 2 life-13-00296-t002:** Strategies to Enhance self-efficacy for better CPAP adherence [34].

Education	Educational material (leaflets, videos) by one-on-one clinic visits, group meetings, telephone calls, telemedicine interactions, official internet sites
Behavioral Interventions	Cognitive behavioral therapy (CBT) Motivational enhancement therapy (MET),
Telemonitoring	Data on treatment effectiveness and level of adherence. Possible mask leaks, residual respiratory events, CPAP use duration

CPAP: Continuous Positive Airway Pressure.

**Table 3 life-13-00296-t003:** Issues that should be discussed during the first visits for CPAP treatment.

Explain about OSA and its impact on patients’ health if left untreated
Suggest lifestyle changes such as weight loss, sleep hygiene
Explaining the importance of treatment with CPAP
CPAP device demonstration: different types of masks, humidifier, ramp
Discuss a follow-up plan (short-term and long-term: face-to-face, telephone, telemedicine)
Solve practical issues with CPAP (see Table 1)

## Data Availability

Data sharing not applicable: No new data were created or analyzed in this study. Data sharing is not applicable to this article.

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
