# Peer review of "Adherence to CPAP Treatment: Can Mindfulness Play a Role?"

_life, 2023, doi:10.3390/life13020296_

Round 1

Reviewer 1 Report

The authors present a review article discussing adherence with CPAP and review the challenges therein.  They discussed several interventions that have been trialed over the years, and propose a mindfulness-based approach that may help.  The authors provide a nice overview of barriers and practical issues as well as proposed solutions.  Table 1 is a nice overview.  Psychological and behavioral barriers are discussed and table 3 provides a summary of suggested interventions.  Finally, the authors discussed the possible role for mindfulness-based approaches, which fit nicely with some of the adherence barriers such as claustrophobia.  They make that a compelling case that this is an important area for future investigation.  Overall, this is a very well written and informative review.

Author Response

We would like to thank the reviewer for the nice comments supporting our manuscript.

Reviewer 2 Report

Overview:

The authors performed a literature review focused on the use of mindfulness to increase compliance with use of CPAP for sleep apnea. As much discussion is devoted to general psychological factors and CPAP compliance, it is unclear why the authors emphasize mindfulness in this review article rather than basing the review on the full spectrum of psychological factors and CPAP compliance. Indeed, since the controlled trials of mindfulness and CPAP compliance remain to be performed, at best the authors are describing the rationale behind a hypothesis.

Lines 58-60:

It has been shown that almost 50% of patients discontinue CPAP after a year of treatment [15-17], within a range of 29% to 83%, while 8 to 15% of patients reject treatment as early as the first night of application [18,19].

Were these studies from North America or did they include patients from other parts of the world. To demonstrate cross-cultural relevance of the authors underlying premise regarding common discontinuance of CPAP and the opportunity for mindfulness to reverse this phenomenon, it would be nice to show that this is a global issue and that culturally sensitive might be used throughout the world. 

Table 1:

I am not sure this summary helps make the case above what is stated in lines 82-84 below. I recommend deletion of Table 1, along with reference in the following lines.

However, some studies have shown that the reduction of side effects with different solutions, as shown in Table 1, did not correlate with increased quality of life and CPAP adherence [16, 17, 33].

Lines 90-95:

Education is recommended by the American Academy of Sleep Medicine as an important component of adherence [35]. However, there is evidence those educational interventions about OSA and its consequences when untreated, on the different types of devices and masks, or on providing solutions to resolve the different problems, did not result in the complete improvement of adherence [16, 17, 31-33].

In these studies, who did the education and was the education in person or by text? Unless a knowledgeable and trusted health professional provides the education regarding the use of the CPAP device and its expected benefits, the impact of education is highly speculative. 

Lines 130-133:

Bodenhausen has examined the role of stereotypes and biases highlighting bias on perceptions of court cases and has found that "morning people" were more likely to engage in stereotyping at night and "night people" were more likely to engage in stereotyping in the morning [42].

Although the above statement is an interesting study finding, its relevance to compliance with the CPAP device remains unclear to this reader. Could the authors be more explicit regarding how such stereotyping might be related to CPAP compliance?

Lines 158-159:

On the other hand, patients with Type D personality are reluctant to follow medical advice and show decreased adherence to CPAP treatment [51].

The authors should define “Type D personality.”

Lines 309-315:

Personalized medicine should be the future target of treatment individualizing adherence goals by treating patient specific symptoms (as excessive daytime sleepiness or insomnia) and reducing the risk of patient specific consequences (as cardiovascular)[3]. Mindfulness can therefore serve as a novel approach to promote CPAP adherence in OSA patients by reducing emotional distress and increasing subjective well-being. Future research should continue to investigate the role of mindfulness-based interventions in CPAP treatment adherence.

As noted at the beginning of this manuscript review, since the controlled trials of mindfulness and CPAP compliance remain to be performed, at best the authors are describing the rationale behind a hypothesis. It may be best to declare this manuscript as a hypothesis focused review justifying future interventional and observational studies on the role of mindfulness as an adjunct to other psychological approaches for enhancing CPAP compliance.

Author Response

Answer to reviewer’s 3 comments

We would like to thank the reviewer for the interesting comments that have improved our manuscript.

  • Overview:

The authors performed a literature review focused on the use of mindfulness to increase compliance with use of CPAP for sleep apnea. As much discussion is devoted to general psychological factors and CPAP compliance, it is unclear why the authors emphasize mindfulness in this review article rather than basing the review on the full spectrum of psychological factors and CPAP compliance. Indeed, since the controlled trials of mindfulness and CPAP compliance remain to be performed, at best the authors are describing the rationale behind a hypothesis.

Answer to reviewer:

We would like to thank the reviewer for the comment. The scope of this review was to investigate if mindfulness may play a supportive role in CPAP adherence of patients with OSA. Mindfullness has been used in other sleep disorders as insomnia. However data on OSA patients are limited and controlled trials of mindfulness and CPAP compliance remain to be performed. We have added this very important point to several points (abstract, introduction, conclusion) in the revised manuscript. On the other hand we believe that throughout the review we have explained how mindfulness may help to increase CPAP adherence.

  • Lines 58-60:

It has been shown that almost 50% of patients discontinue CPAP after a year of treatment [15-17], within a range of 29% to 83%, while 8 to 15% of patients reject treatment as early as the first night of application [18,19].

Were these studies from North America or did they include patients from other parts of the world. To demonstrate cross-cultural relevance of the authors underlying premise regarding common discontinuance of CPAP and the opportunity for mindfulness to reverse this phenomenon, it would be nice to show that this is a global issue and that culturally sensitive might be used throughout the world. 

Answer to reviewer:

Thank you  for your remark as some of the references come from North America. However in the comprehensive systematic literature review of Rotenberg et al [18] data from studies from other parts of the world are included. In this study that evaluated data regarding CPAP adherence over a twenty year timeframe, it was found that CPAP adherence remained persistently low, around 34% (30–40 %). They also found that approximately 11% of the participants of the 82 CPAP trials they evaluated, were unable to remain on CPAP treatment over the duration of the trial.  Discontinuance of CPAP is a global problem despite different cultural characteristics of the patients. Mindfulness is getting more popular during  the last years all over the world and has been used in other sleep disorders as insomnia, , and for that we believe that it would be interesting to use it as an alternative method to improve CPAP adherence.

  • Table 1:

I am not sure this summary helps make the case above what is stated in lines 82-84 below. I recommend deletion of Table 1, along with reference in the following lines.

However, some studies have shown that the reduction of side effects with different solutions, as shown in Table 1, did not correlate with increased quality of life and CPAP adherence [16, 17, 33].

Answer to reviewer

Thank you for your point. As the manuscript is a review we would like to keep Table 1 so that the reader will have at a glance an overview of the most frequent problems during CPAP use and their solutions. Additionally the other reviewer supported this table.

However if you insist, we could have Table 1 in the supplement.

We have deleted as you suggested lines 82-84 and have added the following

‘It has been shown that healthcare professionals have different perceptions and knowledge compared with patients regarding CPAP side effects, possible problems and educational needs. This is important in the design of educational programmes of healthcare professional and patients in order to increase CPAP adherence [33]. The education that is provided by a knowledgeable and trusted health professional regarding the use of the CPAP device and its expected benefits is of a great importance’

  • Lines 90-95:

Education is recommended by the American Academy of Sleep Medicine as an important component of adherence [35]. However, there is evidence those educational interventions about OSA and its consequences when untreated, on the different types of devices and masks, or on providing solutions to resolve the different problems, did not result in the complete improvement of adherence [16, 17, 31-33].

In these studies, who did the education and was the education in person or by text? Unless a knowledgeable and trusted health professional provides the education regarding the use of the CPAP device and its expected benefits, the impact of education is highly speculative.

Answer to reviewer:

As reported in ‘Treatment of Adult Obstructive Sleep Apnea With Positive Airway Pressure: An American Academy of Sleep Medicine Systematic Review, Meta-Analysis,and GRADE Assessment’ (ref 35), ‘The delivery of education varied substantially between studies  and included being given written materials, watching a video, or face-to-face didactic sessions.’’ A clinically significant improvement in PAP adherence was demonstrated in patients who received educational intervention compared to usual care (PAP usage > 4 h/night, OR 1.2 (95% CI: 0.8 to 1.9).

We agree with the reviewer that the education that is provided by a knowledgeable and trusted health professional regarding the use of the CPAP device and its expected benefits is of a great importance. (We have added this comment in the revised manuscript.)

  • Lines 130-133:

Bodenhausen has examined the role of stereotypes and biases highlighting bias on perceptions of court cases and has found that "morning people" were more likely to engage in stereotyping at night and "night people" were more likely to engage in stereotyping in the morning [42].

Although the above statement is an interesting study finding, its relevance to compliance with the CPAP device remains unclear to this reader. Could the authors be more explicit regarding how such stereotyping might be related to CPAP compliance?

Answer to reviewer:

Thank you for your comment. We have added the following in the revised manuscript:

‘ Bodenhausen has examined the role of stereotypes and biases especially in a lack of motivation and has found that "morning people" were more likely to engage in stereotyping at night and "night people" were more likely to engage in stereotyping in the morning [42]. The importance of circadian variations is particularly interesting because it may affect several different processes i.e.  biological, social and cicardian.  The knowledge of this may provide a different perspective on the expectations of CPAP compliance according to circadian variations. It would be rather difficult to expect a ‘night’ person to apply CPAP early at night than a ‘morning’ person.

  • Lines 158-159:

On the other hand, patients with Type D personality are reluctant to follow medical advice and show decreased adherence to CPAP treatment [51].

The authors should define “Type D personality.”

Answer to reviewer:

Thank you for your comment. We have added the definition of type D personality in the revised manuscript. ‘Individuals with type D personality (D stands for distressed) are characterized by increased negative emotions in different situations and social inhibition as they do not share their emotions with others, because they are afraid of possible disapproval or rejection.’

  • Lines 309-315:

Personalized medicine should be the future target of treatment individualizing adherence goals by treating patient specific symptoms (as excessive daytime sleepiness or insomnia) and reducing the risk of patient specific consequences (as cardiovascular)[3]. Mindfulness can therefore serve as a novel approach to promote CPAP adherence in OSA patients by reducing emotional distress and increasing subjective well-being. Future research should continue to investigate the role of mindfulness-based interventions in CPAP treatment adherence.

As noted at the beginning of this manuscript review, since the controlled trials of mindfulness and CPAP compliance remain to be performed, at best the authors are describing the rationale behind a hypothesis. It may be best to declare this manuscript as a hypothesis focused review justifying future interventional and observational studies on the role of mindfulness as an adjunct to other psychological approaches for enhancing CPAP compliance.

Answer to reviewer:

Thank you for this important point. As we have answered at your first comment we agree that there is a lack of controlled trials for the role of mindfulness for CPAP adherence.  This review supports the hypothesis that mindfulness may be used as an alternative method to increase CPAP adherence by reducing the emotional stress and increasing the sense of well being as an adjunct to other psychological approaches.

We have added this comment in the revised manuscript

Round 2

Reviewer 2 Report

I suggest tightening up the literature review a bit to reduce overall manuscript size by 1-2 pages. 

Author Response

We would like to to thank the reviewer for the comments on our manuscript.

I suggest tightening up the literature review a bit to reduce overall manuscript size by 1-2 pages

Answer to the reviewer

We can understand your comment as the literature on mindfulness and OSA is limited. However in the directions for authors, the review articles are required to be around 4000 words at minimum and for that we have done a more extensive review.